# Understanding the Emergence of Rural Agrotourism: A Study of Influential Factors in Jambi Province, Indonesia

Zulgani Zulgani [1,*], Junaidi Junaidi [1], Dwi Hastuti [1], Ernan Rustiadi [2,3], Andrea Emma Pravitasari [2,3] and Fadwa Rhogib Asfahani [1]

[1] Department of Economics, Faculty of Economics and Business, Universitas Jambi, Jambi 36361, Indonesia; junaidi@unja.ac.id (J.J.); dwihastuti@unja.ac.id (D.H.); fadwarhogib@gmail.com (F.R.A.)

[2] Division of Regional Development Planning, Department of Soil Science and Land Resources, Faculty of Agriculture, IPB University, Bogor 16680, Indonesia; ernan@indo.net.id (E.R.); andreaemma@apps.ipb.ac.id (A.E.P.)

[3] Center for Regional System Analysis, Planning, and Development (CRESTPENT), IPB University, Bogor 16127, Indonesia

\* Correspondence: zulgani@unja.ac.id

**Abstract:** This investigation delineates the multi-faceted determinants integral to the evolution of agrotourism within rural domains, concentrating on the province of Jambi as a case study. This scholarly inquiry engaged with four representative villages, utilizing primary data procured through focus group discussions (FGDs) and comprehensive interviews with various stakeholders. These encompass village administration, village-owned enterprises (BUMDes), youth organizations, the regional body for planning and development, the tourism office, community figureheads, village facilitators, commercial operators, and local community delegates. The analytical methodology incorporated the transcription of FGDs and comprehensive interviews, data distillation, analytical interpretation, and triangulation. The NVivo 11 Plus suite facilitated this qualitative data analysis. The investigation discerned six cardinal determinants that substantively influence the trajectory of agrotourism development within rural areas. These include the potential of nature tourism, the accessibility, and caliber of agro-products, the adequacy of infrastructure, the involvement and roles of community and institutional bodies, technological innovation, and the safeguarding of local cultural heritage. This scholarly inquiry underscores the necessity of a collaborative approach in formulating and implementing policies. This approach, which calls for the inclusion of diverse stakeholders, is aimed at bolstering the sustainable development of agrotourism.

**Keywords:** agrotourism; rural; sustainable development

## 1. Introduction

Thus far, urban-centric development has yielded considerable socio-economic and environmental repercussions for rural and urban areas. The prevailing relationship between these areas is often detrimental to rural locales, with cities predominantly serving as conduits for the resources harvested from the countryside (Agunggunanto et al. 2016; Wilonoyudho 2017; Zhe and Ai 2020). This dynamic precipitates deforestation and environmental degradation in rural areas, ultimately engendering poverty among the rural populace (Akpinar et al. 2005; Burki et al. 2021; Malkanthi and Routry 2011).

In light of these issues, the past decade has seen escalating interest in rural development across the globe, particularly in developing nations. Notably, in Indonesia, this trend was underscored by the ratification of Law Number 6 of 2014 about villages (Indonesia 2014). This legislation affected a shift in the strategy for village development, which was previously under the purview of higher administrative echelons such as the regency, municipal, provincial, or national government. The new approach empowers village inhabitants to determine their development model (Pamungkas 2019; Murtadho et al. 2020).

Furthermore, the law assures financial support for village initiatives via fiscal transfers to village funds, colloquially termed Dana Desa (Anshari 2018).

Rural development—a paradigm oriented towards augmenting the well-being of rural communities—entails constructing and enhancing infrastructure, economic systems, and suitable technology (Rustiadi et al. 2018b; Rustiadi et al. 2023). Moreover, it places villagers at the forefront, from planning to field implementation.

Within this context, rural development with an agricultural emphasis is gaining momentum in various regions. However, the large-scale cultivation of the agricultural sector faces obstacles such as limited land tenure and impoverished farmers. Consequently, farming households find themselves obliged to diversify both their on-farm and off-farm production to safeguard the financial stability of their families (Barbieri and Mshenga 2008; Junaidi et al. 2020, 2022; Tamburini et al. 2020)

There is a considerable range of possibilities for off-farm production in rural settings. Rural areas possess an array of intriguing potential for development, particularly those tied to the natural environment's authenticity, diverse agricultural commodities, distinctive customs, arts, and culture, and the immense potential for implementing agrotourism. Each region's unique rural conditions present a compelling allure for tourists. Consequently, the development of agribusiness and rural agrotourism programs has become a strategic government initiative to foster rural areas and enhance the welfare of rural communities.

Tourism, one of the most prominent economic sectors, can primarily drive economic growth (Goodwin and Chaudhary 2017; Gunarta and Hanggara 2018; Tabash et al. 2023). Agrotourism, a subset of agriculture-based tourism, offers potential solutions to a myriad of socio-economic challenges in rural communities, thereby acting as a catalyst for rural economic development and growth (Chen and Diao 2022; He et al. 2022; Lak and Khairabadi 2022; Obeidat and Hamadneh 2022). The development of rural and agrotourism can positively impact business success and aid in diversifying business risks (Mura and Ključnikov 2018; Petrović et al. 2017; Rosalina et al. 2021; Cheteni and Umejesi 2023). Furthermore, agrotourism development can contribute to sustainable development and environmental preservation (Fafurida et al. 2023; Vysochan et al. 2022).

In Indonesia, the tourism sector holds a significant role in boosting foreign exchange revenues. This is demonstrated by the considerable contribution of the tourism sector to foreign exchange earnings, which is approximately USD 4.26 billion (Widi 2022), or around 3.10 percent of the total foreign exchange in 2022, amounting to USD 137.2 billion (BPS 2022e). Tourism is the fourth-largest contributor after oil and natural gas, coal, and palm oil exports.

Jambi Province in Indonesia has diverse agrotourism potentials within its villages (Rustiadi et al. 2023; Kusumastut and Mukhzarudfa 2018; Wahyuni and Syamsir 2021). Agrotourism and nature-based tourism activities involving residents' participation are increasingly vibrant in the rural areas of Jambi Province (Zuriati and Mariya 2020; Saadah et al. 2021). Within the sustainable development paradigm framework, an apt strategy for developing a tourist village is the implementation of community-based tourism (Giampiccoli et al. 2020; Manaf et al. 2018; Demkova et al. 2022). This strategy hinges on community empowerment, sustainability, conservation, and cultural enhancement, to improve the residents' livelihoods (Giampiccoli and Kalis 2012; Song et al. 2021; Jin et al. 2022). Community-based tourism strategies have been deployed for tourism development in several developing Asian countries (Rocharungsat 2008). However, the support from local government and surrounding communities for developing community-based agrotourism is deemed suboptimal. Thus, there is a call for enhanced synergy between the two, accompanied by various realistic development support recommendations (Andayani et al. 2022).

Developing agrotourism villages in Jambi Province presents an intriguing study area, particularly concerning the factors influencing these villages' evolution. This research is anticipated to lay the groundwork for future policy formulation for rural area development.

Many studies have explored the factors influencing the development of agrotourism villages. These factors can essentially be categorized into the following: (1) environmental factors, such as ecological landscape and diversity of agricultural and plantation products

(Lengkong et al. 2018; Rodrigues et al. 2006); (2) socio-economic factors, including social kinship, social capital, community involvement and participation, financial and human resources, cultural landscape, marketing, and institutions (Anita 2017; Dayan and Sari 2022; Faganel 2011; Hrymak et al. 2019; McGehee and Kim 2004; Ohe and Kurihara 2013); (3) supporting factors, such as government policies and other stakeholders (Rustiadi et al. 2018a), encompassing planning, legalization, provision of assistance, and others (Tew and Barbieri 2012; Schilling et al. 2012; Nana 2020; Saraswati et al. 2020); and (4) social and technical infrastructure factors (Agafonova and Spektor 2023; Anugraheni and Astutiningsih 2021; Baranova and Kegeyan 2019; Busby and Rendle 2000; Ćirić et al. 2021; Evgrafova and Ismailova 2021; Kachniewska 2015; Nickerson et al. 2001).

Previous research in this domain has indisputably established a robust foundation. However, most prior studies primarily investigated single or multiple locations sharing similar agrotourism characteristics. This approach resulted in individual studies identifying different and location-specific factors, leading to a relatively wide variation in identified elements. The absence of a common factor influencing the development of agrotourism villages inevitably hinders the formulation of effective policies applicable across diverse agrotourism village characteristics.

This study diverges from its predecessors by exploring several distinct locations rather than concentrating on a single site or multiple sites with similar agrotourism traits. The research scrutinizes four various agrotourism sites: those founded on forest resources, coastal and mangrove environments, horticultural plants, and tea plantations. The study aims to identify common determinants by examining these diverse agrotourism areas, which can be broadly applied across various agrotourism locations.

Theoretically, this approach is expected to find common factors influencing the development of agrotourism villages across varied village characteristics. Furthermore, building on this foundation of common factors, this study aims to practically contribute to formulating policies for developing agrotourism villages. The policies generally apply across various village characteristics, enhancing their effectiveness and sustainability.

## 2. Methodology

### 2.1. Research Locations

This research was conducted in Jambi Province, Indonesia, focusing on four villages chosen as study samples: (1) Tanjung Lanjut Village, Muaro Jambi District; (2) Kuala Lagan Village, East Tanjung Jabung Regency; (3) Renah Alai Village, Merangin District; and (4) Mekar Sari Village, Kerinci Regency (Figure 1). These villages were selected based on their significant agrotourism potential and ongoing development as tourist destinations, specifically agrotourism, within Jambi Province.

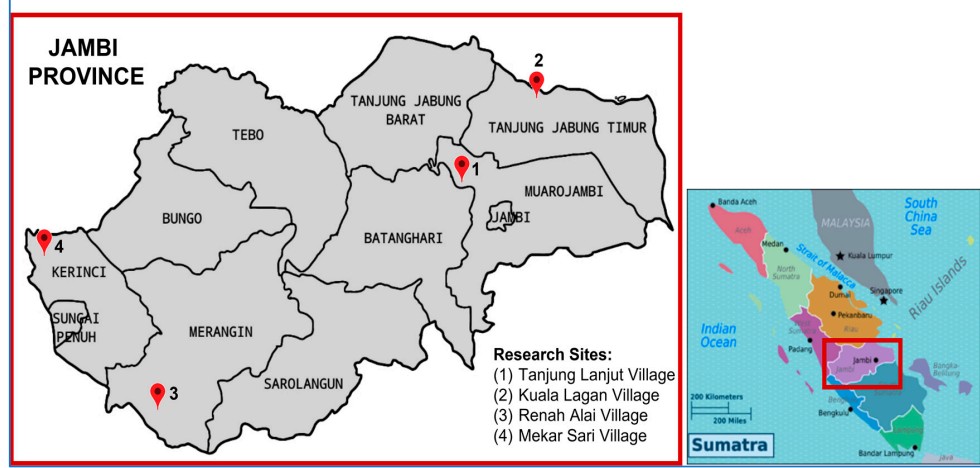

**Figure 1.** Research Locations. Source: https://en.wikivoyage.org/wiki/Sumatra (accessed on 10 June 2023) (Sumatra Map) and layout by the author (Jambi Province Map).

*2.2. Data Collection*

Primary data were collected through focus group discussions (FGD) and in-depth interviews. The focus group discussion (FGD) activities were conducted four times, with each instance taking place in each research village. The number of FGD participants in each village amounted to seven individuals. FGD participants comprised stakeholders, including village officials, village-owned enterprises (BUMDes), youth organizations, the regional body for planning and development (Bappeda), the tourism office, village facilitators, and commercial operators. The FGD topics were related to the factors influencing the development of agrotourism villages, including both driving and inhibiting factors, challenges, and prospects. Subsequently, a systems-based approach was employed to identify relationships or the risk impact on the entire system and to make appropriate strategic decisions to manage such risks.

Following the FGD implementation, in-depth interviews were conducted with community figureheads and two local community delegates in each village. The interview questions aimed to delve deeper into the findings obtained from the FGD activities.

*2.3. Data Analysis*

The data were analyzed using a qualitative approach. Through this qualitative approach, it is hoped that the research can reveal an in-depth, detailed, and natural understanding of agrotourism in rural areas and provide an opportunity to uncover new findings or themes unrecognized or unmeasurable by quantitative methods.

The data analysis process involved recording the results of FGDs and in-depth interviews, performing data reduction, analyzing and interpreting the data, and conducting triangulation. Data reduction, an ongoing activity throughout data collection, involved simplifying, selecting, and transforming raw data from field notes. This step included identifying themes or patterns and coding the data meaningfully.

In order to ensure the validity of the data, various triangulation methods were employed in this study: method triangulation (using FGD and in-depth interviews), inter-researcher triangulation (verification by other researchers), data source triangulation (utilizing information from various stakeholders, community figureheads, and local community delegates), and theory triangulation (applying different theoretical perspectives in data analysis).

This study utilized NVivo 11 Plus software to facilitate the qualitative data analysis. The text data accumulated from focus group discussions (FGDs) and meticulously transcribed in-depth interviews form the bedrock of the data analysis process. Upon importing these data into the NVivo software, the researchers construct nodes for the concepts emerging from the data and initiate the coding process, which entails marking sections of the text corresponding to specific nodes.

Subsequently, in order to undertake a more comprehensive analysis of the data, a series of approaches are adopted, including "word frequency", which gauges the prevalence of certain terms; "text search", enabling a detailed examination of the text; "matrix coding query", which aids in the exploration of potential relationships within the data; "cluster analysis", which categorizes similar entities into groups; and "content analysis", a technique for making replicable and valid inferences from the data.

*2.4. Sample Village Profile*

The following section provides a concise profile of the four sampled villages, the information for which is derived from observational studies, interviews conducted with the village administrative staff, and available documents within the village governance.

2.4.1. Tanjung Lanjut Village

Tanjung Lanjut is one of the villages located in Muaro Jambi Regency. The distance of this village from the regency capital is approximately 26.8 km, and the journey can typically be covered in about 40 min. Most Tanjung Lanjut Village residents are primarily oil palm or rubber farmers or employees of oil palm companies in partnership with the village (BPS 2022d).

Tanjung Lanjut Village, which was recognized as a Tourism Village in 2021, possesses considerable agrotourism potential. Since 2015, this village has been home to a village-owned enterprise (BUMDes) named Tanjung Jaya Mandiri (Tajam). This BUMDes manages Lake Tangkas, a tourist attraction that spans approximately 413 hectares. The administration and development of this lake by BUMDes commenced in 2018.

Tanjung Lanjut Village offers more than just nature tourism, which has started to draw a growing number of visitors from inside and outside Jambi Province. It also presents a unique potential for agrotourism development. One of its main features is the Pendant Forest, a forest situated on the periphery of Lake Tangkas, which can be explored by boat. This forest is renowned for its predominance of Putat trees, which the local community has begun to exploit to produce Putat tea. The journey through the Pendant Forest is part of a village tour package provided by BUMDes, including activities such as boat tours, banana boat rides, and duck encounters.

### 2.4.2. Kuala Lagan Village

Kuala Lagan is a village situated in the Tanjung Jabung Timur Regency. The distance from this village to the regency's capital is around 63.0 km, and the travel time to reach the destination is typically around two hours. The primary livelihood of the residents of Kuala Lagan Village involves managing coconut, areca nut, and oil palm plantations (BPS 2022c).

Kuala Lagan Village is the only village in East Tanjung Jabung Regency that boasts a 100-hectare Mangrove Forest in the Water Resources Development and Conservation (PKSDA) area. This Mangrove Forest yields natural resources such as forest honey, giant prawns, and crabs. The development of mangrove forests for agrotourism began in 2021.

Since 2016, this village has had a village-owned enterprise (BUMDes). Although it only started actively supporting tourism in early 2021, it has provided tour boat facilities to navigate tourists through the mangrove forest. During specific seasons, the BUMDes also offer tourists educational tours featuring lobster and crab-catching activities. The attraction of lobster and crab catching represents the most appealing activity for tourists. Furthermore, with the increasing influx of tourists, there has been a simultaneous development of culinary tourism independently managed by the local community, particularly focused on seafood-based cuisine.

### 2.4.3. Renah Alai Village

Renah Alai Village, situated in the Jangkat District of Merangin Regency, is rich in agricultural assets and nature-based tourism potential. The distance from this village to the regency capital is approximately 105 km, with a typical travel time of around three hours. The majority of the local population is involved in horticulture and gardening, producing superior commodities such as potatoes, chilies, cabbage, beans, oranges, strawberries, cinnamon, sweet potatoes, and coffee (BPS 2022a). A strawberry-picking tourism experience already exists, managed by the local community. Additionally, the community has processed a portion of the harvested crops into souvenir products for tourists, such as potato chips, sweet potato chips, and coffee powder.

In the Mount Masurai region, Renah Alai Village has geographical conditions highly conducive to agriculture and agrotourism. In addition, the village possesses nature tourism potential, including waterfalls and customary forests, which are still in the developmental planning stage. Despite being in the development phase, the location has already attracted many tourists. The management of this tourist site remains under the stewardship of the local village government.

Recognized as a hub for producing agricultural commodities, Renah Alai Village is a bustling center of activity. Given its cool climate, owing to its location in the mountainous Masurai Valley, this village has the potential to produce staples such as rice, coffee, corn, cassava, strawberries, potatoes, chilies, cabbage, and others.

Renah Alai Village, also lauded as the most beautiful village in 2017–2018, has adorned its streets with vibrant henna flowers. Moreover, the village is known as "Rampai Masurai" due to the beauty and diversity of its agricultural products.

### 2.4.4. Mekar Sari Village

Mekar Sari Village is one of the villages situated in the Kerinci Regency. The distance from this village to the regency's capital is approximately 31.0 km, with an estimated travel time of around one hour. Most Mekarsari Village residents depend on farming and gardening as their primary source of income (BPS 2022b).

According to the Ministry of Tourism, Mekarsari Village, acknowledged as a tourist village, has reached an advanced stage of tourism development. Strategically situated on a hill and near the Kayu Aro Tea Plantation and other natural attractions, this village serves as a tourist-stop destination. The village has also developed a variety of tourist attractions managed by village-owned enterprises (BUMDes), such as a strawberry farm, homestays with panoramic views of the Kayu Aro Tea Gardens and Mount Kerinci, and cafes offering a modern ambiance within the homestay area.

The village's main products include agricultural, plantation, and forest commodities such as potatoes, cinnamon, coffee, strawberries, and forest honey. Through BUMDes Mekar Sari, the village has developed several business units, including strawberry picking tours at the Bukit Cinta homestay, a cafe with captivating natural views, a cattle farm, and a savings and loan cooperative that supports small businesses in Mekarsari Village. Nearly every home in this village operates a small business, a shop, or agro-product production. As such, the development of agrotourism has been successful in Mekarsari Village.

## 3. Results and Discussion

Six internal factors have been identified as influencing agrotourism development in the studied villages. These six internal factors include the following: (1) cultural heritage; (2) community and institutions; (3) nature tourism potential; (4) technological innovation; (5) infrastructure; (6) agro-products. These factors, presented in Figure 2, provide a comprehensive overview of the various determinants that influence the development and success of agrotourism in the studied villages. Understanding these factors can provide valuable insights for other villages and regions seeking to develop agrotourism initiatives.

The internal factors impacting agrotourism development are as follows: (1) Cultural heritage: This non-material aspect represents the identity of a society or nation passed down from previous generations and preserved for the present and future ones. Cultural heritage can manifest in objects, sites, or cultural values inherited from the past (Díaz-Andreu 2017; Holtorf 2018). (2) Community and institutions: This factor includes the role of various formal institutions within the village and the broader community, including organizations or community groups that support these institutions (Beckmann et al. 2021; High et al. 2005). (3) Nature tourism potential: This factor involves leveraging the potential of resources and environmental management (Xu et al. 2023). Nature tourism can also harness the potential of the environment and its resources, both in their natural state and combined with human-made elements (Hashemkhani et al. 2015; Falk et al. 2022). (4) Technological innovation: This refers to discovering something new or modern. In agrotourism, technological innovation involves using technology to market or showcase the agrotourism potential of an area (Madanaguli et al. 2022; Roman et al. 2020). (5) Infrastructure: Infrastructure involves the presence and quality of necessary physical and social facilities, like roads, bridges, and irrigation systems, which are needed for community activities and businesses (Buhr 2003). (6) Agro-Products: These are any raw or processed agricultural commodities or products, including those from livestock or those sold in the market for human consumption (Dos-Santos 2020).

Understanding these factors is crucial for crafting strategies to develop and promote agrotourism in a given location. This multi-faceted approach ensures a holistic understanding of the various elements to consider when planning agrotourism development.

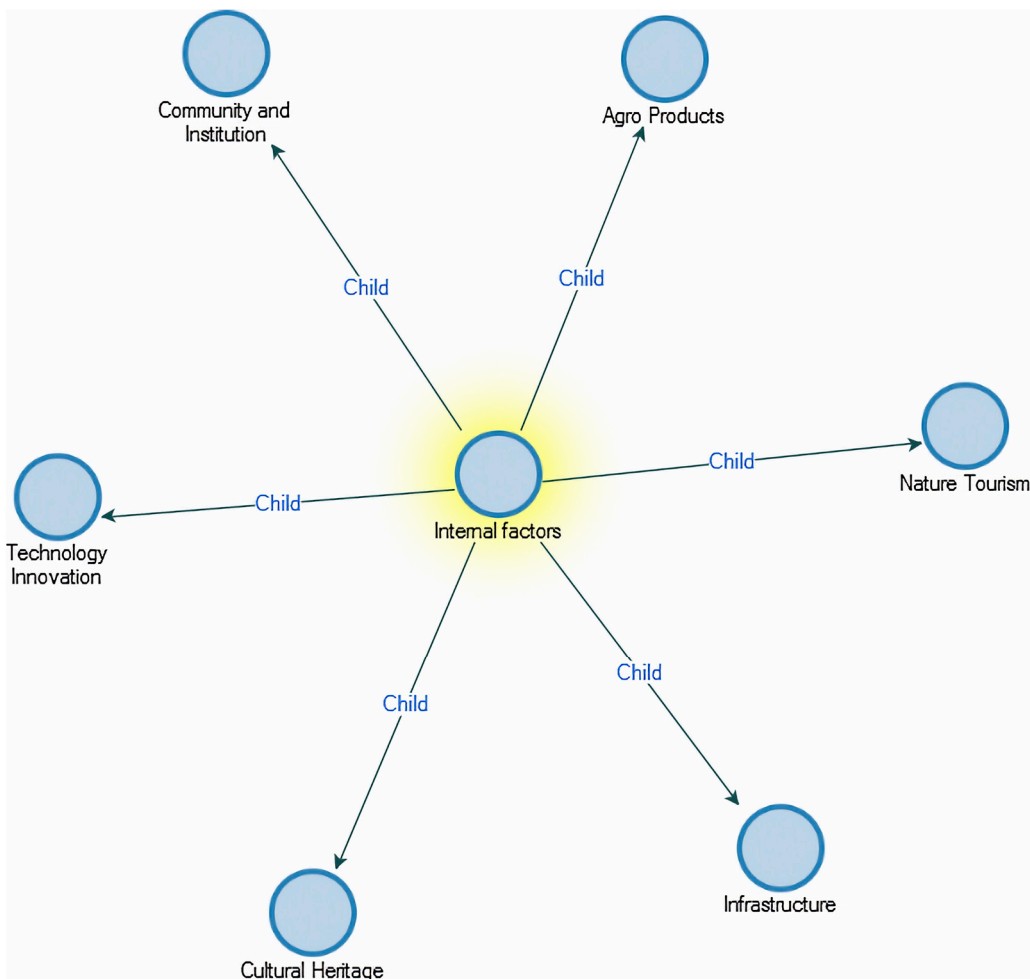

**Figure 2.** Exploring the determinant factor diagram for agrotourism development.

Based on the six factors identified in the four sample villages, a pairwise comparison was conducted between factors and villages. Figure 3 provides a comparison between agro-products and infrastructure as internal factors.

Agro-products include the development of various agricultural commodities and various supporting factors related to the development of agro-products. These sub-factors include culinary agro-products, lack of business networks, financing or capital, human resource development, and agro-product potential. On the other hand, infrastructure focuses on developing facilities and infrastructure that support agrotourism development. Sub-factors in infrastructure include access to village roads, access to tourist roads, homestays, security, tourism infrastructure, and culinary and souvenir centers.

One of the community leaders in Mekar Sari Village said: "When you look at our village in terms of agrotourism and natural beauty, we're like a bright shining star with all kinds of farm produce. Visitors can even pick strawberries while they soak in the Kayu Aro Tea Garden views on one side, and Mount Kerinci on the other. Regarding tourist facilities and infrastructure, we've got some stuff in place, like homestays, a cafe, ATV rides, and even a flying fox. But we still don't have a place where folks can buy souvenirs. This can be a bit of a headache for visitors who want to take back a keepsake from our village. Plus, our homestays could use sprucing up to make tourists feel more at home."

The residents in Renah Alai Village share the same concerns about the lack of a souvenir center and the need for more homestays. As one villager stated: "We're planning to propose to the government for funds to set up more homestays and food outlets. We want to create a food and souvenir center so visitors can take back a piece of our village, like our famous Jodah Bakar".

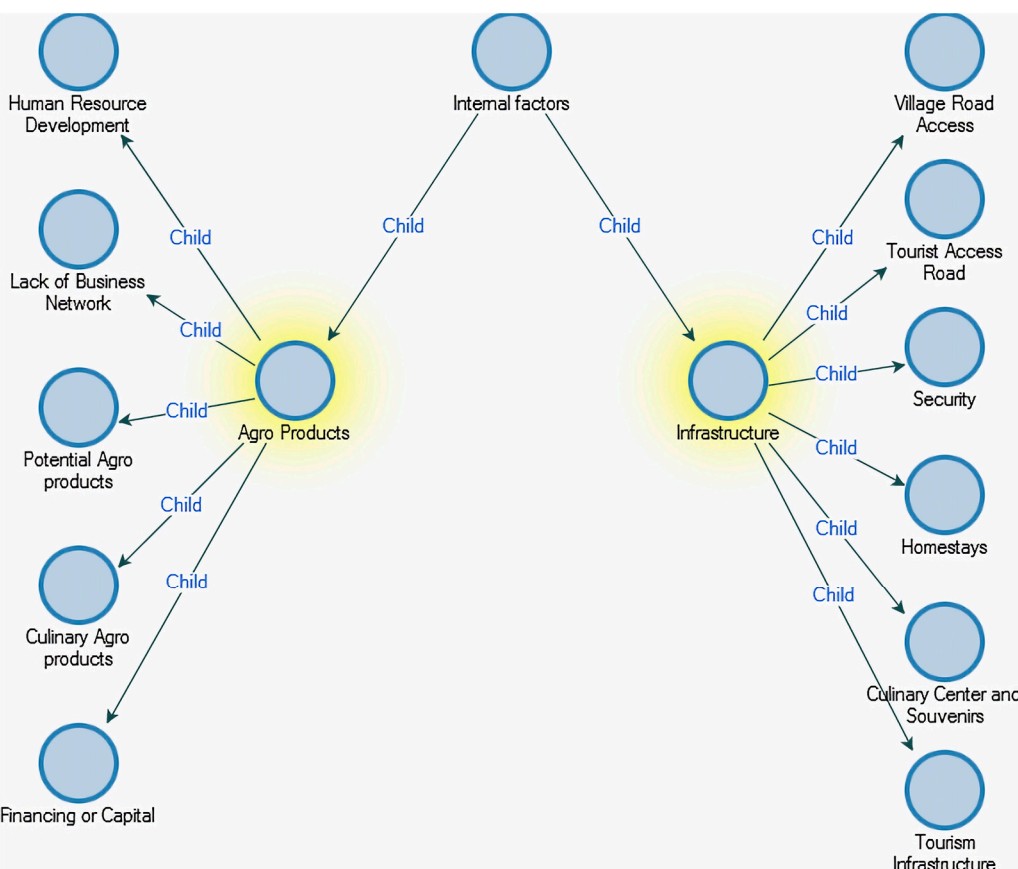

**Figure 3.** Comparison of agro-products and infrastructure.

Figure 4 illustrates the relationship between nature tourism potential, cultural heritage, customary law, cultural traditions, and culinary traditions. Intriguingly, this relationship was only identified in three study villages: Tanjung Lanjut, Renah Alai, and Mekarsari. In contrast, this relationship was not identified in Kuala Lagan Village.

The interplay between nature tourism potential and cultural heritage is vital in developing agrotourism. Natural attractions, such as Tangkas Lake in Tanjung Bawah Village, the waterfall in Renah Alai Village, and Mount Kerinci and horticulture in Mekar Sari Village, draw visitors to these areas. However, the cultural heritage of each area, such as customary laws, cultural traditions, and culinary traditions, often provides a unique identity and appeal to these locations.

A notable person in Mekar Sari Village stated that Renah Alai Village has a fair share of unique attractions. One local we conversed with shared: "We've got Pencak Silat and local traditional dances. Not to mention, our local grub has got the nod from the ministry, no less. Our Jodah Bakar recently bagged a world record at the Jangkat festival for being the biggest of its kind. Jodah, now that's a sweet thing that we put inside bamboo. Once it's dried out, we slice and roast it. But we can really show off here this thing called Lukah Gilo. It's like a shell made from bamboo that's got a skull on it, complete with eyes and all decked out in clothes. After some mystical chanting, four blokes can hardly hold it back."

These cultural elements can deeply enrich a visitor's experience, allowing them to learn about and appreciate these areas' unique customs, traditions, and cuisines. This can greatly enhance the attractiveness of these agrotourism destinations and can even serve as a unique selling point.

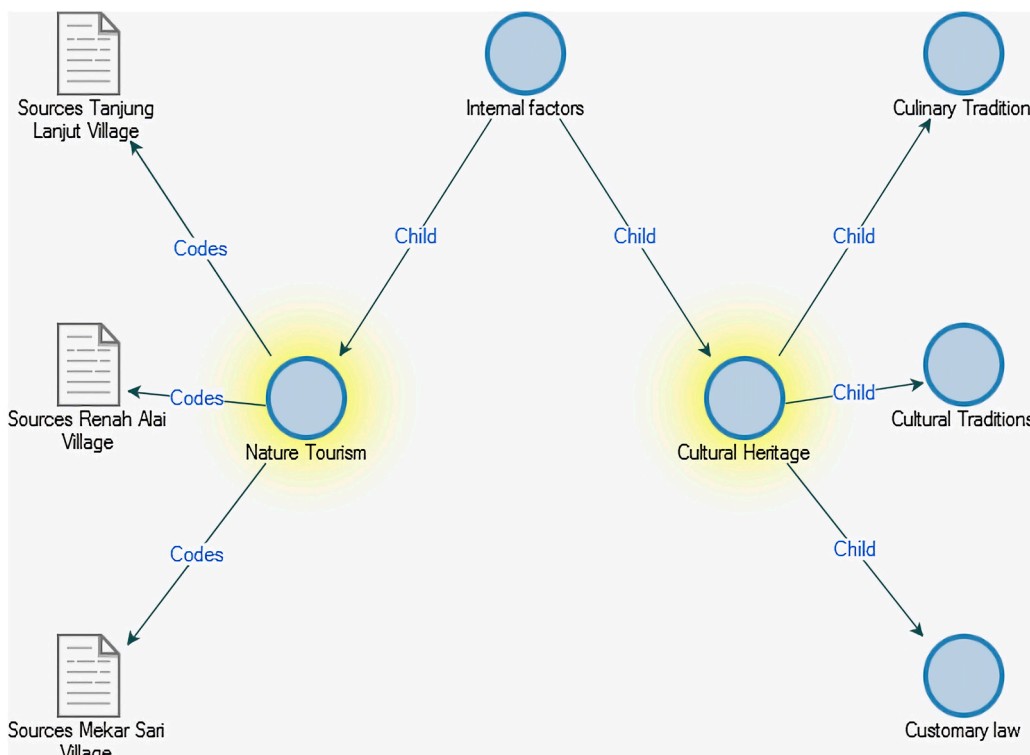

**Figure 4.** Comparison of nature tourism potential and cultural heritage.

It is essential, however, that the development of agrotourism in these areas remains sustainable. This means ensuring that using natural resources and promoting cultural heritage does not harm the environment or undermine the cultural integrity of these areas. As such, it is crucial to promote environmental and cultural education among the public, implement effective marketing strategies, and encourage the active participation of local communities.

This way, agrotourism development can promote these areas as attractive tourist destinations and contribute to their social, economic, and environmental sustainability. This aligns with the broader goal of achieving sustainable development, which seeks to meet the needs of the present without compromising the ability of future generations to meet their own needs.

Moreover, Figure 5 delineates the relationship between technological innovation and community and institutions.

Figure 5 illustrates that community and institutional factors significantly influence agrotourism development. They provide support and direction, foster collaboration, and enhance the visibility and marketability of agrotourism initiatives. Village-owned enterprises (BUMDes), universities, marketing agencies, and the local government, among others, are all crucial actors in this regard. They can facilitate agrotourism development by providing resources, expertise, and promotional opportunities.

For example, BUMDes can promote and manage local agrotourism attractions, while universities can provide research support and technical assistance. Marketing agencies can help improve the visibility and marketability of agrotourism destinations, and local governments can provide support through policy and regulatory measures.

When it comes to the role of the community, one of the community leaders from Tanjung Lanjut Village said: "Our nature tourism spot, Danau Tangkas, is something we've been managing since 2018. Since that year, we've really buckled down with the community's support because we probably wouldn't be where we are today without them. Right from the get-go, we came together in a communal effort introduced and integrated into our village government program by our village head. It's included in our RPJMDes that rolls out every six years".

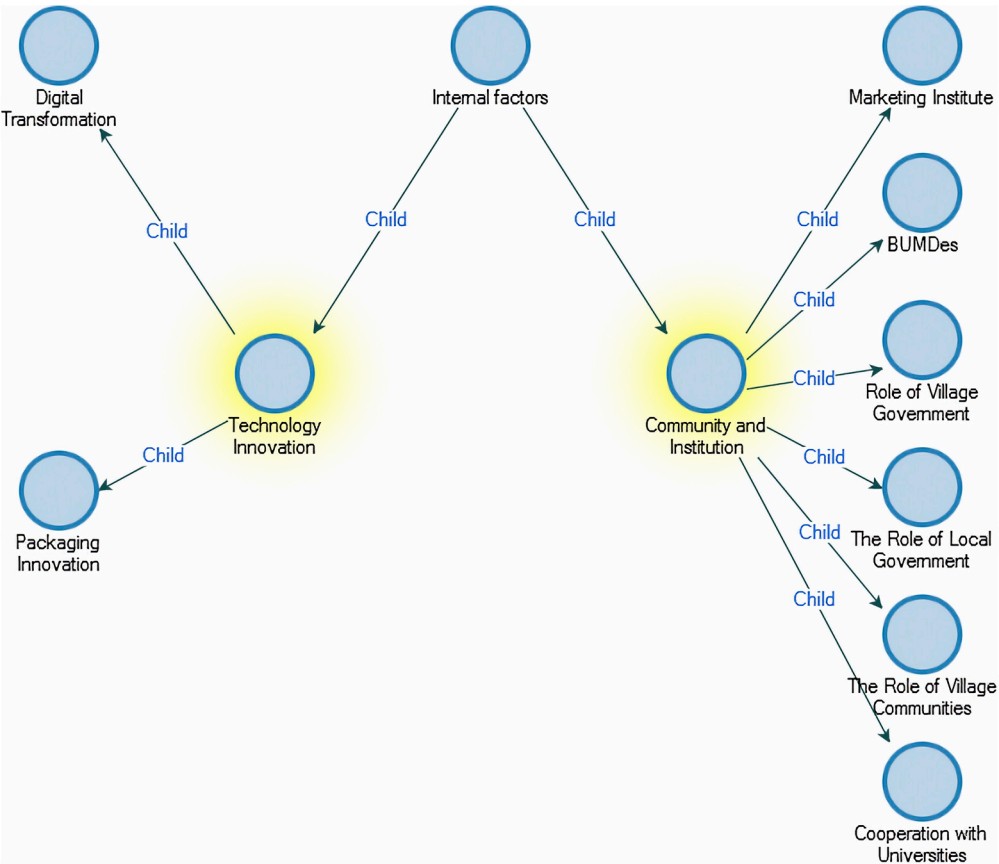

**Figure 5.** Comparison of innovation and community and institutions.

As for the role of universities, a community figure from Kuala Lagan Village shared: "Not many folks in our community are familiar with the term 'agrotourism', and even fewer understand the impact it can have on our village. However, with partnerships we've got going with Jambi University and others, like the National KKN program for example, at the very least, we're getting a sense of how our mangrove forests can help boost our village's income. We're hoping that developing our mangrove forests can create job opportunities, especially for our folks still looking for work".

Incorporating technological innovation into these efforts can further enhance the effectiveness and reach of agrotourism initiatives. Packaging innovation can add value to agrotourism products, making them more attractive and marketable, while digital transformation can increase operational efficiency and broaden market reach. Community and institutional actors can promote and support these technological innovations and provide education and training on using these technologies.

Figure 6 represents the amalgamation of Tanjung Lanjut Village and Kuala Lagan Village sources. There are 13 sub-factors in Tanjung Lanjut Village and 10 in Kuala Lagan Village. Each sub-factor is interrelated between the Tanjung Lanjut Village and the Kuala Lagan Village.

Each village's unique environment and resources necessitate tailored strategies for agrotourism development. For both Tanjung Lanjut Village and Kuala Lagan Village, common strategies include the development of human resources, enhancing the quality and variety of agricultural products, improving village road accessibility, bolstering village-owned enterprises (BUMDes), enhancing the role of local government, providing financial or capital support, and fostering cooperation with universities. These strategies aim to enhance local communities' skills and knowledge, infrastructure, institutional support, and the overall quality of agrotourism offerings.

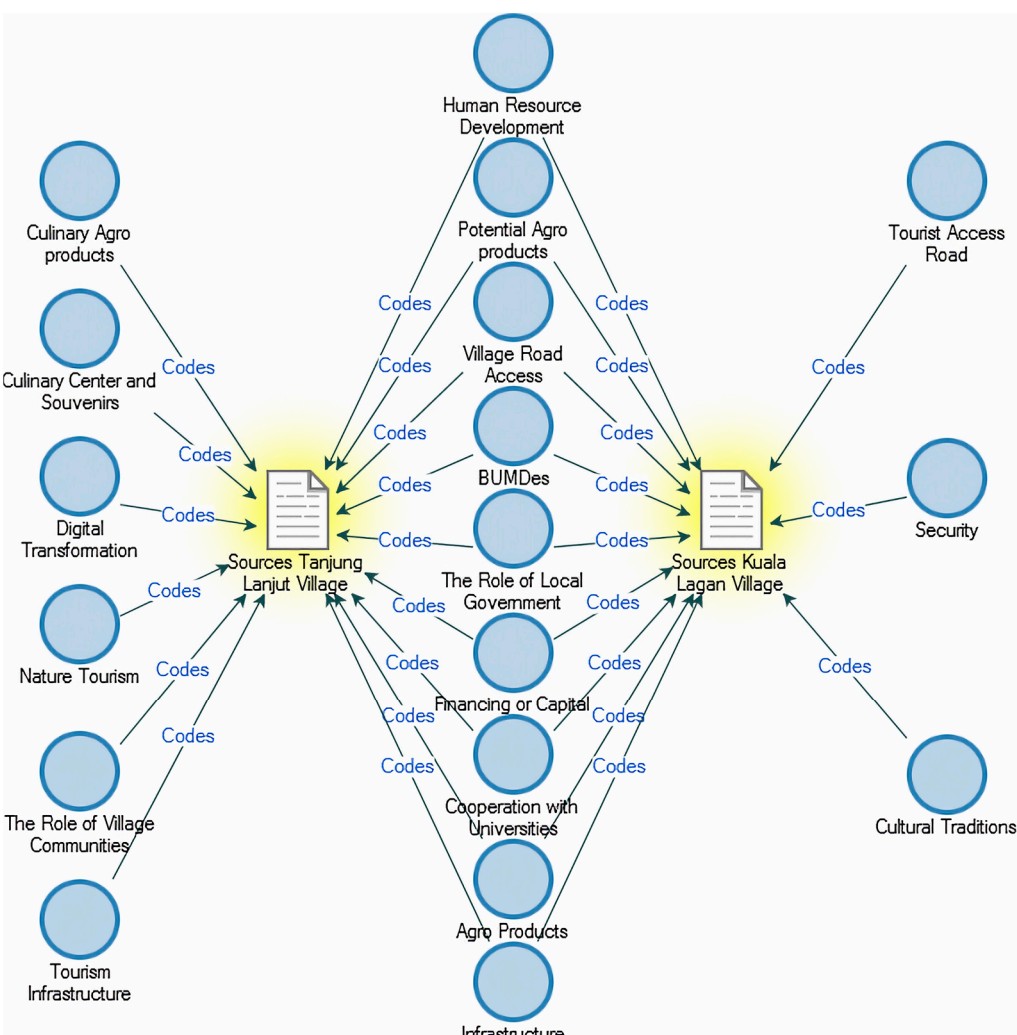

**Figure 6.** Comparison of Tanjung Lanjut Village and Kuala Lagan Village.

According to a local we conversed with in Kuala Lagan Village: "We've had our village-owned enterprise, or BUMDES as we call it, since 2016, but it's only this year, in 2022, that it's really got its gears grinding to support our tourism activities. And the folks running the BUMDES? They're the university students we've got in our village. Turns out, these youngsters have got plenty of bright ideas up their sleeves".

However, each village has unique strategies based on individual characteristics and resources. In Tanjung Lanjut Village, the focus is on developing culinary agro-products, culinary centers, and souvenirs, leveraging digital transformation, harnessing the potential of nature tourism, enhancing the role of the village community, and improving tourism infrastructure. These strategies underscore the village's focus on culinary development, digital technology use, and tourism infrastructure maintenance and development.

An elder from Tanjung Lanjut Village shared that "Our little corner of the world, it sits right on edge (where the districts of Muaro Jambi and Batanghari meet). Once upon a time, folks from the outside would say our village was where 'ghosts dumped their kids' because no outsiders ever touched it and it was so darn quiet. Nobody, not even officials, would've wanted to step foot in our village unless there was something special about it. The creation of Danau Tangkas tourism got the wheels of government programs rolling into our village. We started seeing road paving, social aid, and all that jazz. And that keeps my spirits high, pitching in and helping develop village tourism. It's a pride of its own to see my village go viral and get known by folks from outside. We've got the Putat trees, which only grow around the lake. They've become part of the tourist attraction since visitors can

walk amongst these Putat trees in the forests lining the lake. We've also started processing the Putat leaves, drying them out to make tea. The combination of the beauty of Danau Tangkas, the Putat trees, Liontin flowers, treehouse, water tourism, bridges, homestays, MSMEs, and Putat tea products makes Tanjung Lanjut Village an interesting and unique tourist destination that folks of all ages want to visit".

In contrast, Kuala Lagan Village emphasizes improving road access, enhancing safety measures, and preserving and utilizing cultural traditions as a tourist attraction. These strategies reveal the village's priority on infrastructure enhancement, tourist safety, and celebrating and utilizing cultural traditions to attract tourists.

As shared by a local leader from Kuala Lagan Village, in a tone more fitting of a rustic villager: "We've got a fair share of tourist attractions, no denying that. We got ourselves a mangrove forest, we got a culture that's worth a look. The trouble is, we ain't been able to make good use of 'em. Few folks feel like heading this way because the roads are in such a bad state".

Figure 7 displays the fusion of sources from the Tanjung Lanjut Village and Renah Alai Village. There are 13 sub-factors for Tanjung Lanjut Village and 10 sub-factors for Renah Alai Village. Of the various sub-factors, eight sub-factors are interrelated between the two villages.

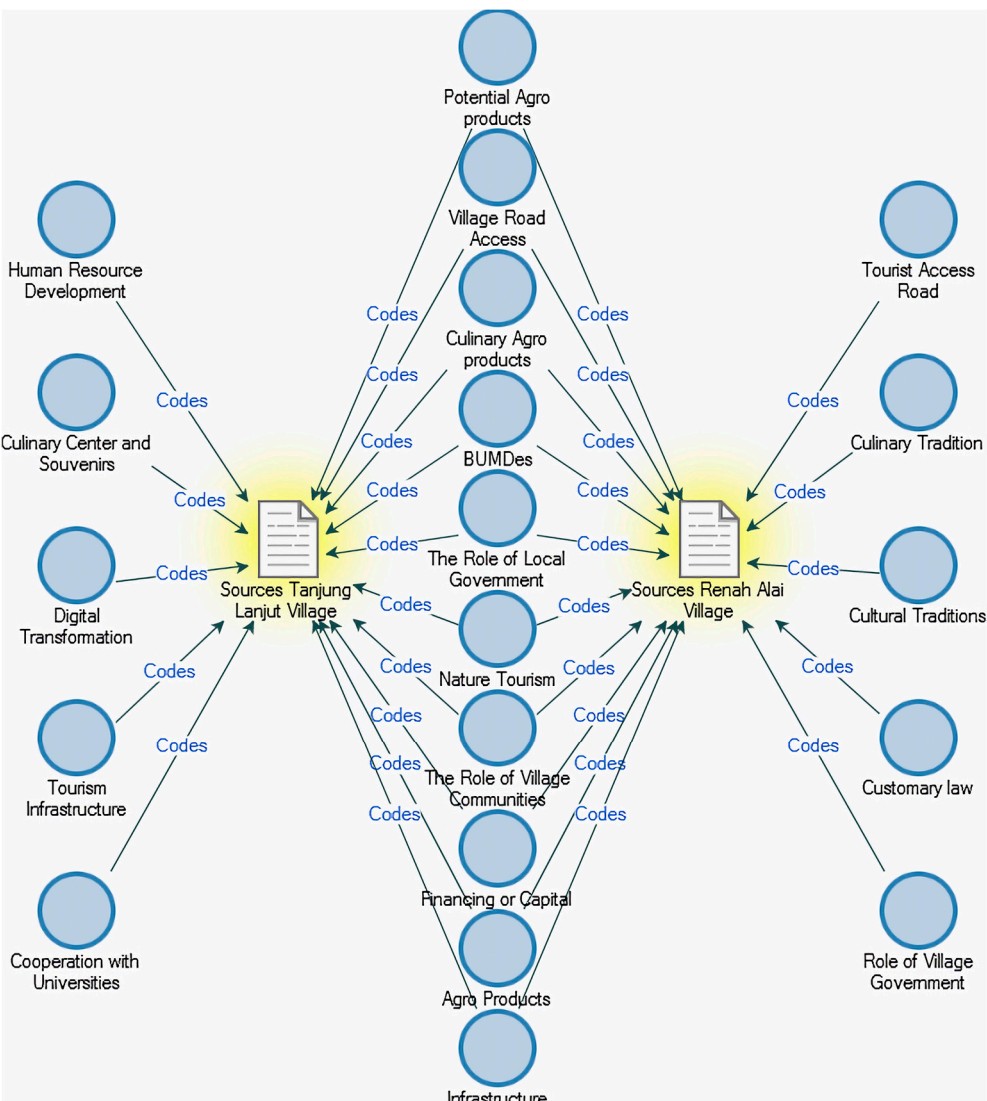

**Figure 7.** Comparison of Tanjung Lanjut Village and Renah Alai Village.

Both Tanjung Lanjut Village and Renah Alai Village share several strategic sub-factors in agrotourism development, which include enhancing agro-product potential, improving village road accessibility, promoting culinary agro-products, bolstering village-owned enterprises (BUMDes), strengthening the role of local government, developing nature tourism potential, enhancing the involvement of village communities, and increasing financing or capital. These shared strategies aim to maximize the use of local resources, enhance community and government participation, and foster a strong foundation for agrotourism development.

Regarding how the folks of the village are involved, essentially, they approve of the development of agrotourism and are open to investors and any developmental programs the government is planning. As a community leader in Renah Alai stated, in a more folksy vernacular: "We got plenty of land, and our folks are open and ready to welcome investors or any government programs aimed at using our land to develop tourist attractions".

However, each village also employs unique strategies tailored to its specific circumstances and resources. In Renah Alai Village, these unique strategies involve improving access to tourist roads, preserving and promoting culinary and cultural traditions, respecting customary law, and strengthening the role of the village government. These strategies signify a commitment to local cultural preservation, enhancement of tourism infrastructure, and strengthening local governance supporting agrotourism development.

As expressed by the folks of Renah Alai, in a more down-to-earth manner: "We're still dealing with problems like broken roads, a lack of transportation, scant tourist facilities and bad internet service, all of which make tourists less keen to come here. If we're looking to develop tourism in the future, the key is the road, seeing as a lot of our economic activity is also alongside the road".

Conversely, Tanjung Lanjut Village emphasizes human resource development, culinary and souvenir centers, digital transformation, tourism infrastructure, and collaboration with universities. These unique strategies highlight the village's commitment to enhancing human resource quality, leveraging digital technology in agrotourism development, improving tourism infrastructure, and enhancing community capacity and knowledge through partnerships with higher educational institutions.

Figure 8 combines Tanjung Lanjut Village and Mekar Sari Village sources. There are 21 sub-factors in the two villages, of which 6 are only in Tanjung Lanjut Village, and 8 are present in Mekasari Village, with 7 sub-factors in both villages.

Tanjung Lanjut Village and Mekarsari Village share seven key potential sub-factors in developing agrotourism. These shared strategic sub-factors are as follows: agro-product potential, improving access to village roads, promoting culinary agro-products, strengthening the role of local government, developing nature tourism potential, increasing financing or capital, and developing tourism infrastructure. These shared strategies highlight a mutual commitment to maximize local potential and create an environment that supports agrotourism development.

However, each village has unique strategic sub-factors tailored to their specific circumstances and resources. For Mekarsari Village, the focus is on expanding business networks, developing marketing institutions, improving access to tourist roads, developing homestays, preserving and promoting culinary traditions, safeguarding cultural heritage and customary law, and innovating product packaging. These unique strategies underscore Mekarsari Village's emphasis on marketing, infrastructure development, cultural preservation, and product innovation.

Moreover, Tanjung Lanjut Village focuses on developing human resources, strengthening village-owned enterprises (BUMDes), creating culinary and souvenir centers, implementing digital transformation, enhancing the role of village communities, and collaborating with universities. These unique strategies highlight Tanjung Lanjut Village's commitment to human resource development, digital technology adoption, and collaboration with educational institutions for agrotourism development.

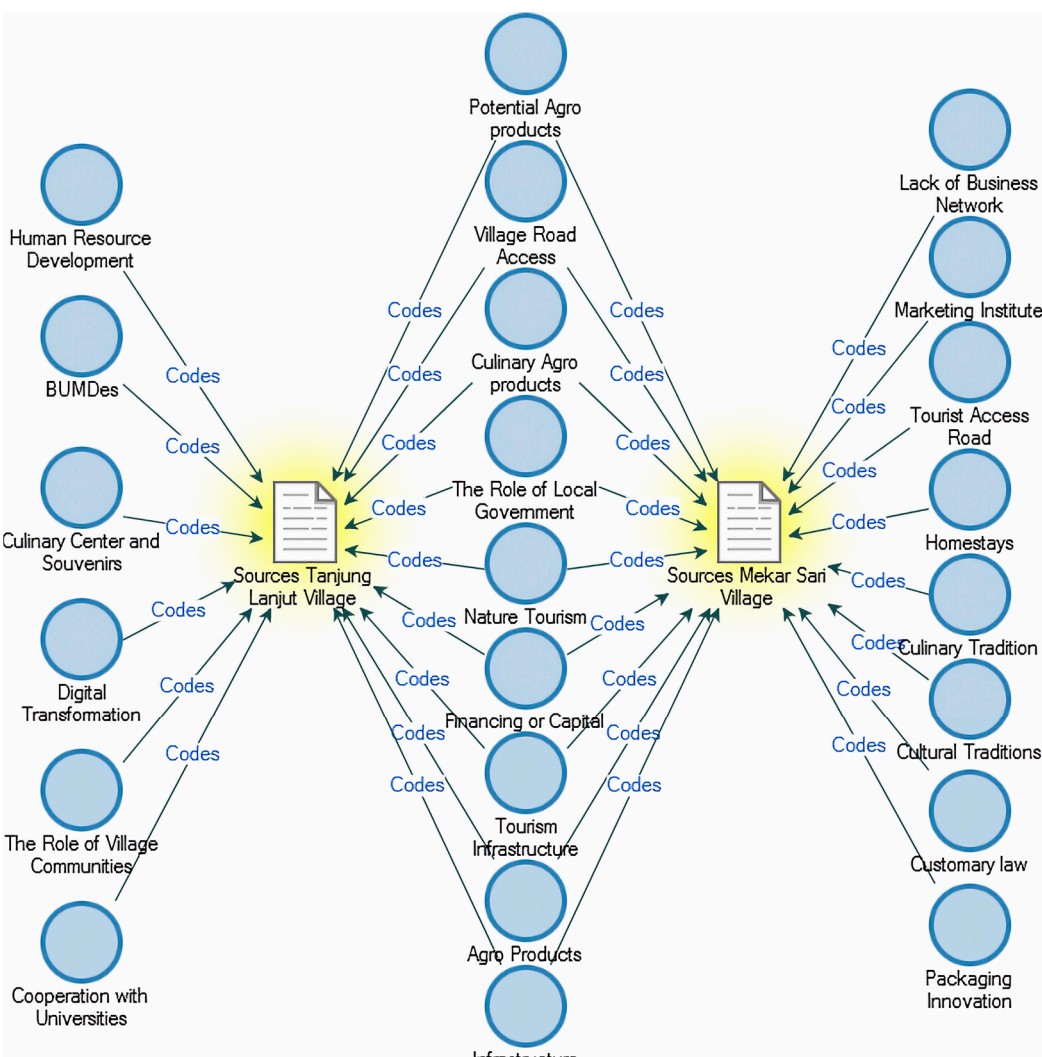

**Figure 8.** Comparison of Tanjung Lanjut Village and Mekarsari Village.

Figure 9 portrays the amalgamation of Kuala Lagan Village and Renah Alai Village sources. There are 16 sub-factors in both villages, of which 3 are only present in Kuala Lagan Village, 6 are only present in Renah Alai Village, and 7 sub-factors are present in both villages.

Kuala Lagan Village and Renah Alai Village share several internal sub-factors in their approach to agrotourism development. These shared strategic sub-factors include the following: agro-product potential, improving access to the village and tourist roads, preserving cultural traditions, developing village-owned enterprises (BUMDes), strengthening the role of local government, and increasing financing or capital. These shared strategies highlight key focus areas for both villages in their pursuit of agrotourism development.

However, each village also has unique strategic sub-factors tailored to their specific circumstances and resources. For Kuala Lagan Village, the focus is on developing human resources, enhancing security, and establishing collaborations with universities. These unique strategies suggest that Kuala Lagan Village prioritizes improving the quality of its human resources, ensuring a safe and comfortable environment for tourists, and leveraging the knowledge and skills that can be gained through collaborations with educational institutions.

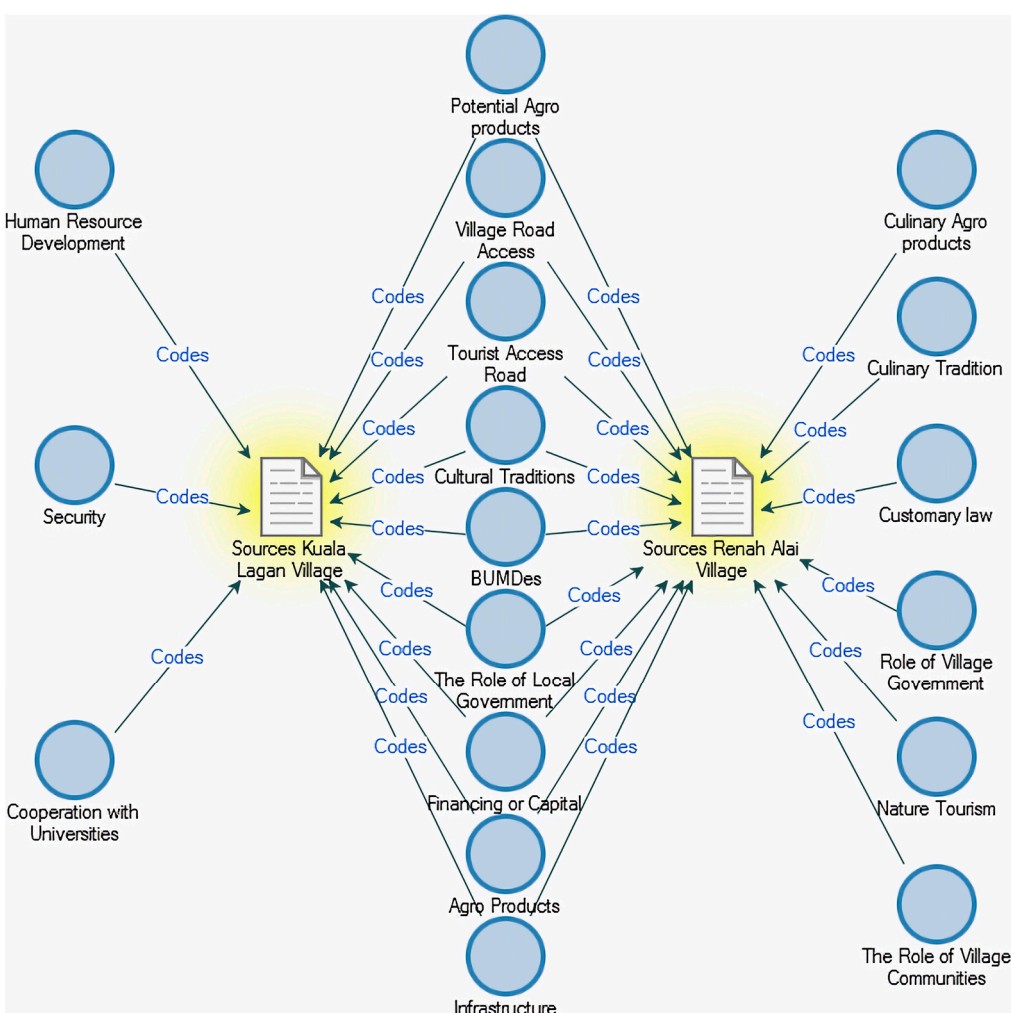

**Figure 9.** Comparison of Kuala Lagan Village and Renah Alai Village.

On the other hand, Renah Alai Village focuses on developing culinary agro-products, preserving and promoting culinary traditions, preserving customary law, strengthening the role of the village government, developing nature tourism potential, and enhancing the role of village communities. These unique strategies reflect Renah Alai Village's commitment to preserving local culture and traditions and encouraging the active participation of the community and village government in agrotourism development.

Figure 10 exhibits an amalgamation of Renah Alai Village and Mekar Sari Village sources. There are 17 sub-factors in both villages, of which 3 are only present in Renah Alai Village, 5 are only in Mekar Sari Village, and 10 are in both villages.

Renah Alai Village and Mekarsari Village have several shared internal sub-factors in their approach to agrotourism development. These common strategic elements include the following: developing agro-product potential, improving access to village roads and tourist spots, promoting culinary agro-products, preserving cultural traditions and customary law, strengthening the role of local government, enhancing nature tourism potential, and increasing financing or capital. These shared strategies represent vital elements that can greatly impact the progress and success of agrotourism development.

Speaking more informally, one of the locals said: "There's a lot in this village that needs fixing before we can really get tourism up and running. The biggest issue is the access road to the village. It's still pretty treacherous, and no one's got round to paving it yet. If we had a decent road in Mekarsari, it'd be much easier for folks to come and go, and we'd likely get a lot more visitors here".

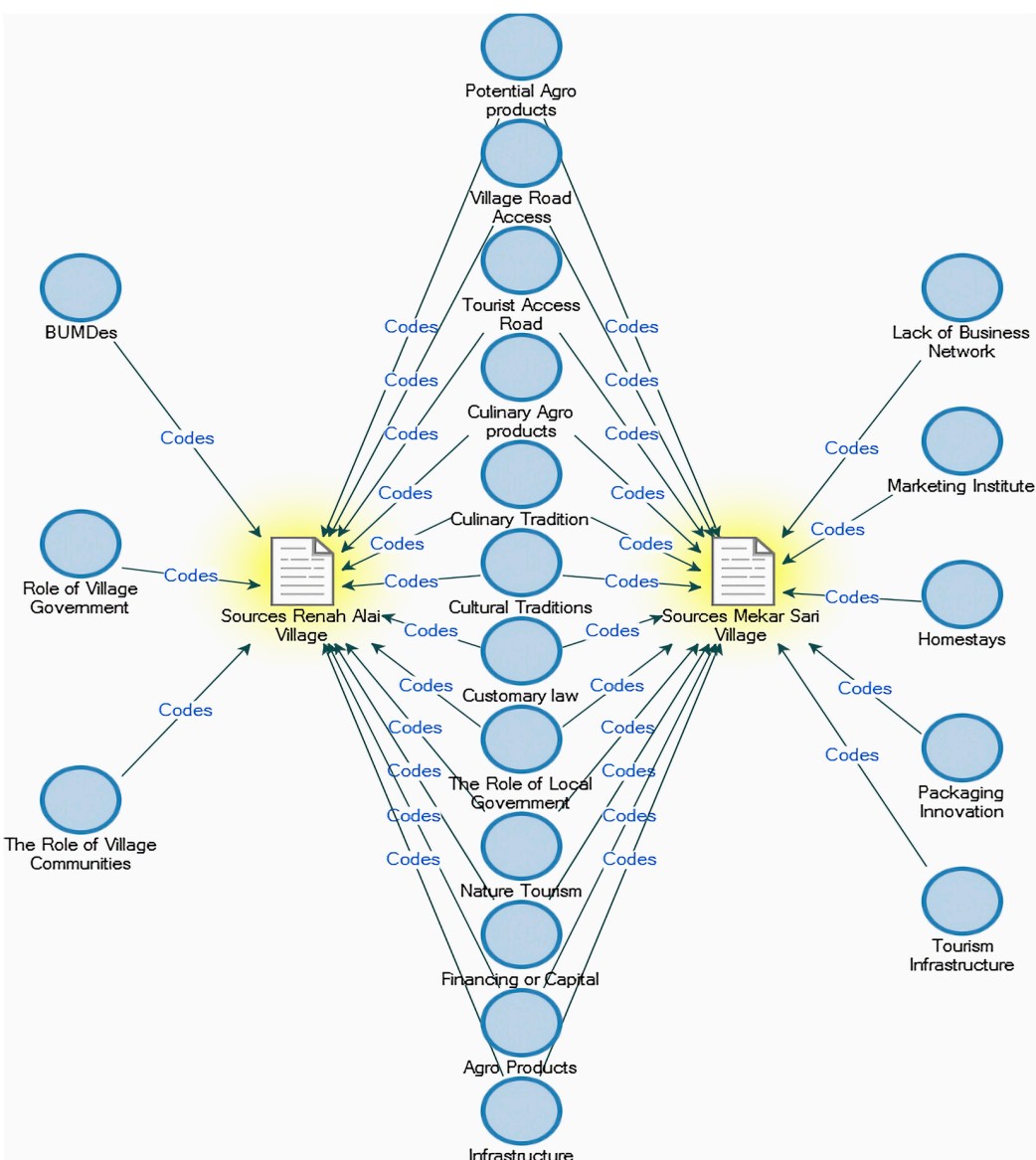

**Figure 10.** Comparison of Renah Alai Village and Mekarsari Village.

However, each village also has unique strategic sub-factors that cater to their specific circumstances and resources. In Renah Alai Village, the primary focus is on developing village-owned enterprises (BUMDes) and enhancing the role of the village government and the village community. This reflects a community-based approach to agrotourism development, which emphasizes the involvement and empowerment of local communities and village governance structures.

On the other hand, Mekarsari Village prioritizes the establishment of business networks, the development of marketing institutions, the provision of homestays, packaging innovation, and the improvement of tourism infrastructure. These unique strategies reveal Mekarsari Village's ambition to explore and leverage new channels and technologies to promote and strengthen its agrotourism offerings.

By understanding and comparing these internal factors, we can identify the key determinants influencing agrotourism development in each village. This is crucial as it allows each village to design and implement the most effective strategies catering to their potential and needs, fostering sustainable and tailored agrotourism development. Furthermore, visualization of internal factor data in the development of agrotourism in each village can provide a clearer and more in-depth picture (Figure 11).

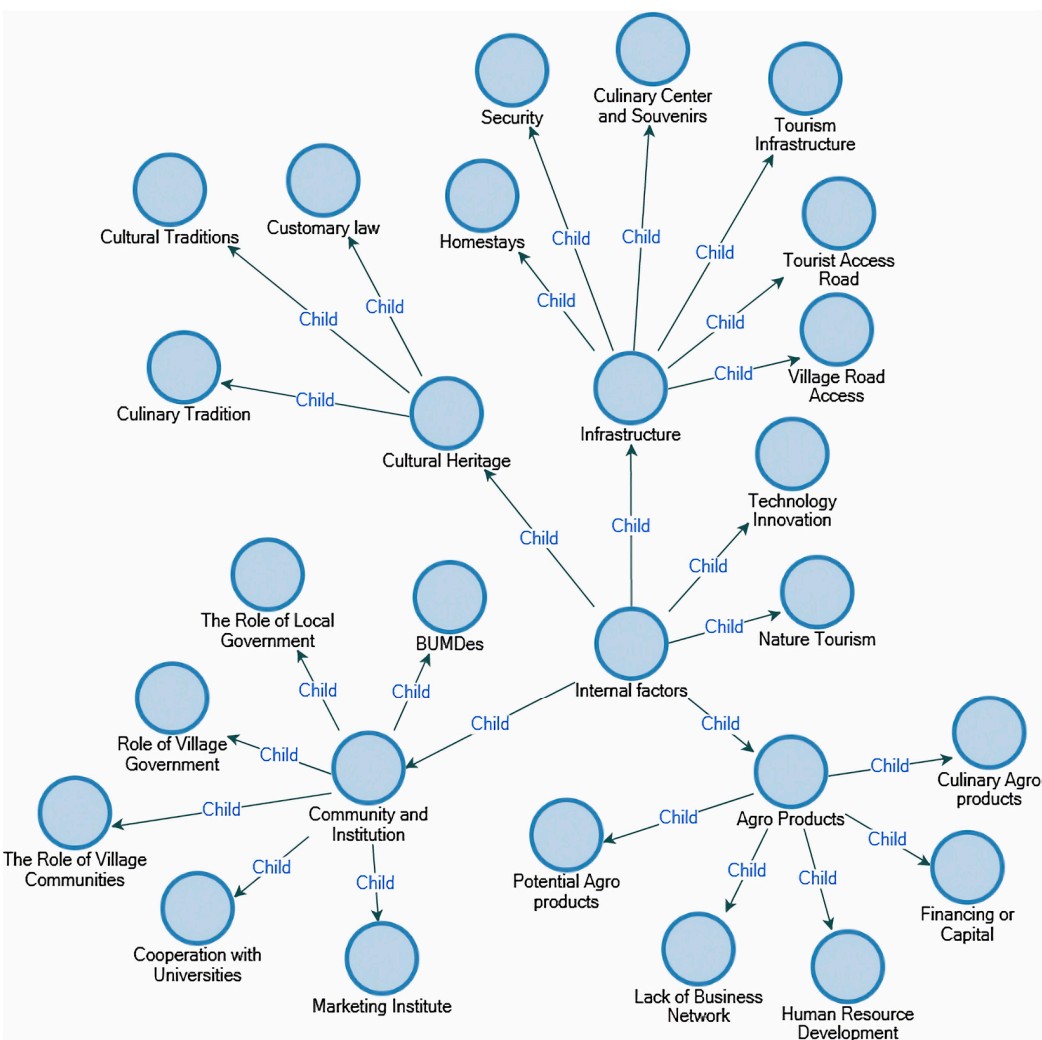

**Figure 11.** Project map for Jambi Province agrotourism development.

The project map detailed in Figure 10 reveals six internal factors that significantly influence agrotourism development at the village level. These factors are nature tourism, agro-products, infrastructure, community and institutions, technological innovation, and cultural heritage.

Based on the "word frequency" analysis, these factors can be arranged as provided in Table 1.

**Table 1.** Hierarchy of factors influencing the development of agrotourism villages in Jambi Province.

| Hierarchy | Tanjung Lanjut | Kuala Lagan | Renah Alai | Mekarsari | Total |
|---|---|---|---|---|---|
| 1 | Nature tourism potential | Nature tourism potential | Nature tourism potential | Nature tourism potential | Nature tourism potential |
| 2 | Agro-products | Heritage | Heritage | Agro-products | Agro-products |
| 3 | Heritage | Agro-products | Agro-products | Heritage | Heritage |
| 4 | Community and Institution | Infrastructure | Infrastructure | Infrastructure | Infrastructure |
| 5 | Infrastructure | Community and Institution | Community and Institution | Community and Institution | Community and Institution |
| 6 | Technology Innovation | | | Technology Innovation | Technology Innovation |

The first factor is the potential of nature tourism, which has emerged as the primary factor across all the investigated villages. The second factor pertains to agro-products. This factor ranks second in two villages, Tanjung Lanjut and Mekar Sari, while it stands third in the remaining two. The third factor is heritage. This factor is the third most influential in two villages, namely Kuala Lagan and Renah Alai, but falls to the fourth position in the other two. Infrastructure serves as the fourth factor, and community and institution, the fifth, influence agrotourist village development. This pattern is found consistently in three villages: Kuala Lagan, Renah Alai, and Mekar Sari. The sixth factor is technology innovation, which is only present in two villages, Tanjung Lanjut and Mekar Sari.

Each factor encompasses several sub-factors that represent unique aspects of agro-tourism development. Except for nature tourism, which in and of itself is considered representative of the village's tourism potential, each factor contains multiple sub-factors.

In the case of the agro-products factor, several potential sub-factors are identified, including agro-product potential, culinary agro-products, the development of human resources managing agro-products, the need for financing and capital, and the lack of business networks marketing agro-products.

A local beekeeper in Mekar Sari village said, "Our honey has this unique sweetness to it, different from any other honey. You see, it comes from bees that feed on cinnamon tree bark. But the thing is, our honey never sells out. We don't really know where to sell it. Plus, we don't know how to package our product".

Aspects such as agro-product potential and culinary agro-products suggest that local agricultural products can serve as major tourist attractions, offering unique added value not available elsewhere. Villages can optimize this potential by investing in human resources development to manage agro-products and ensure their high quality. However, successfully realizing this potential also depends on adequate financing, capital, and a business network that can facilitate product marketing. Without these resources, agro-products may struggle to reach broader markets and generate substantial income. Research by Kharishvili et al. (2019) demonstrates that the effective development of agro-products and related sectors can yield significant economic benefits, create business and job opportunities, and stimulate local revenue growth. Thus, strategic development of the agro-products factor can be a key to successful agrotourism.

From the infrastructure perspective, several sub-factors represent the requirements for agrotourism development. These include access to village roads, which serve as primary avenues for activity mobility within the village, access to tourist sites, tourism-supporting infrastructure, the presence of culinary centers or souvenir shops, the availability of lodgings or homestays, and safety at tourist locations (Tonny and Putri 2020; Zaitul et al. 2022). This illustrates the importance of a well-developed and maintained infrastructure network in promoting and supporting agrotourism activities.

The availability and quality of facilities are crucial determinants of agrotourism's success. This highlights the importance of investing in basic infrastructure and services such as accommodation, transport, and sanitation. Studies by Jaunis et al. (2022), Liew and Ong (2018), Szymanska (2022), and Vengesayi et al. (2009) corroborate this view, underlining that superior facilities are key to the success of a tourist destination and contribute to the allure and competitiveness of these destinations.

Regrettably, some rural areas continue to grapple with infrastructure challenges and underdeveloped basic services, as Mansor et al. (2015) and Azhar et al. (2020) noted. Unpleasant situations like broken toilet doors, uncomfortable buses, and unstable roads can mar the tourist experience and diminish the destination's appeal. Limited public transportation and poorly maintained amenities also pose significant challenges. Therefore, the improvement and maintenance of facilities should be prioritized in agrotourism development, not only to enhance the tourist experience but also to ensure the competitiveness and sustainability of agrotourism in the long term.

Tourist accommodation, including culinary centers and souvenir shops, plays a crucial role in agrotourism development. As Kunáková et al. (2016) and Huang et al. (2023)

described, rural tourism caters to those interested in the rural environment, encompassing a return to nature and a range of accommodation options provided by local families, rural houses, and various other types of countryside lodging.

This implies that developing suitable, high-quality accommodation in rural contexts is vital. These places must offer an authentic and distinctive experience that mirrors the uniqueness and richness of local culture. Additionally, culinary centers and souvenir shops are also important as they allow tourists to taste and take home a piece of local culture.

Nevertheless, achieving this requires a competent and motivated workforce, as Strielkowski (2017) and Vovk and Vovk (2017) emphasize. This workforce must possess the skills and knowledge to provide excellent service and be motivated to perform their best. They should also comprehend the importance of preserving and promoting local heritage and culture. Consequently, human resource development is critical to agrotourism development (Hu et al. 2022).

The third internal aspect, or factor, pertains to the sub-factors representing community and institutional aspects within the village or collaborations implemented in the village. These include the role of the village government in planning village development, the role of the village community in supporting the development of village agrotourism, the role of local government in village development, the activity of BUMDes as village tourism managers, collaborations with universities (particularly in the field of village agrotourism development innovation from a scientific standpoint), and marketing institutions within the village.

In the context of developing agrotourism, the active participation of the local community is a pivotal element (Moise et al. 2023; Nastiti et al. 2019). Obstacles such as constrained capital, limited financial capacity, and knowledge gaps can effectively serve as significant barriers for farmers and local communities to participate (Sipatau et al. 2020; Kolawole et al. 2023). In addition to these challenges, public knowledge deficiencies can also lead to reduced community participation in agrotourism development. In order to counteract this, an integrated approach involving education and enlightenment about the potential benefits of marrying tourism and agriculture is required. This understanding can help farmers and local communities identify opportunities for value-addition from their existing resources, thereby fostering their engagement in agrotourism development (Man and Aspany 2020).

Such efforts to enhance understanding should incorporate various stakeholders, including government entities and educational institutions (Reed 2008). The training and education must be context-specific and delivered in a format and language understandable to the community. In addition, financial backing should also be extended to assist farmers and local communities in surmounting capital and financial constraints. This multi-pronged strategy can bolster local community participation in agrotourism development, contributing substantially to its success and long-term viability.

As the primary authority, local governments promote agrotourism development and stimulate active community involvement in establishing agrotourism enterprises (Yang 2012; Roslina et al. 2022). Governments have the potential to enhance citizens' welfare through various mechanisms, such as extending financial support, strategizing development plans, executing marketing campaigns, and orchestrating training programs, as research by Srisomyong and Meyer (2015) and Chatzitheodoridis and Kontogeorgos (2020) indicates. As per Kubickova and Campbell (2020), the government's role should ideally be centered around policy formulation that aids agrotourism growth, leading marketing and promotional efforts, facilitating access to financial opportunities, disseminating relevant information, and fostering infrastructure development.

However, the role of government cannot stand in isolation. Business entities and other stakeholders also hold significant importance in advancing tourism development (Gajdošík et al. 2018; Lee and Syah 2018). For instance, (Çavusoglu et al. 2020) stress that achieving success in tourism development requires a clear understanding of the government's role as a regulator and business entities as the executors of tourism activities.

In essence, agrotourism development necessitates close collaboration between the government and businesses. The government must foster a conducive environment for agrotourism growth, while businesses need to comprehend and leverage their role in materializing this goal. This success will not only influence economic growth but also enhance the welfare of local communities.

Further, the technological innovation factor indicates the village's ongoing technological shift and its utilization in the context of agrotourism development. Sub-factors representing technological innovation include digital transformation, specifically information and communication technology (ICT) and the internet and social media usage for promoting rural tourism potential (Kumar and Shekar 2020). Additionally, innovation in agro-product packaging is another sub-factor, showcasing the application of modern technology in packaging agrotourism products.

Aligning with the perspective of Linnenluecke et al. (2013), innovation is a fundamental driver in developing tourism products. Innovation assists in adapting to dynamic market demands, ensuring profitability, enhancing competitiveness, and aiding the development of a sustainable tourism industry. In this context, it is important to underscore that technological innovation serves as a tool to advertise and market agrotourism and as a mechanism to augment operational efficiency and effectiveness and enhance the quality of products and services offered. For instance, innovations in agro-product packaging can improve product longevity and quality, while ICT can broaden the market reach and expedite transaction processes. Thus, technological innovation can contribute significantly to elevating the appeal and competitiveness of agrotourism in villages.

The final factor, cultural heritage, encapsulates several sub-factors, including cultural traditions, customary laws, and culinary traditions. The prevailing literature (referenced earlier) has underscored that culture determines tourist attraction and visitor experience.

Each village possesses a unique cultural heritage, which is preserved and passed down by the local inhabitants. This cultural heritage can be village-specific culinary delights, traditional cultural performances, or customary laws enforced by traditional institutions or village elders. Customary law is the rules and norms that villagers and visiting tourists must abide by.

However, it is crucial to understand that cultural heritage is not merely a tourist attraction. Cultural heritage also represents local communities' identity and pride, contributing to agrotourism's value and sustainability (Altassan 2023). Moreover, cultural heritage can serve as an effective instrument for education and learning for visitors, offering a deeper understanding of local communities' histories, traditions, and ways of life. Consequently, preserving and promoting cultural heritage must be executed with respect and maintaining the integrity of local culture while ensuring economic and social benefits for the community.

## 4. Conclusions

The development of agrotourism is a complex and multidimensional process requiring the coordination and cooperation of various stakeholders, including local governments, communities, and business actors.

The development of agrotourism is a multi-faceted and intricate process necessitating coordination and collaboration among a range of stakeholders, encompassing local governments, communities, and business entities. This research identifies six common factors that impact the evolution of villages, namely, nature tourism potential, agro-products, heritage, infrastructure, community and institution, and technology innovation.

The first factor, nature tourism potential, indicates the importance of a region's natural attributes and assets. These could include unique landscapes, flora and fauna, or other natural features that draw tourists. Agro-products, the second factor, refer to the agricultural goods and experiences that tourists can offer. This could range from traditional farming practices and local crops to culinary experiences and locally made products.

Heritage, the third factor, underscores the cultural significance of an area and its people. This could involve historical sites, traditional arts and crafts, folklore, festivals, and

any aspect that reflects the community's heritage. Infrastructure, as the fourth factor, is crucial as it includes not just transportation and accessibility but also the availability of suitable accommodation, amenities, and other services necessary for tourism.

The fifth factor, community and institution, underscores the importance of social structures, local governance, and community engagement in successful agrotourism. A supportive local community and effective institutions can greatly facilitate agrotourism initiatives. Finally, the sixth factor, technology innovation, indicates the increasingly important role that modern technologies, such as digital marketing, online booking systems, and other innovative farming or tourism-related technologies, play in contemporary agrotourism development.

External factors, such as government support and preservation of cultural heritage, also play a significant role in agrotourism development. Local governments have a strategic role in creating a conducive environment for agrotourism growth. At the same time, preserving cultural heritage boosts tourist attraction and represents local communities' identity and pride.

However, challenges remain, including capital and financial constraints and a lack of knowledge and understanding of agrotourism potential. An integrated approach involving education, training, and financial support is needed to address these challenges.

This research provides a more comprehensive understanding of the factors influencing agrotourism development, presenting valuable insights for policymakers, practitioners, and researchers. Governments at all levels—village, district, and provincial—are urged to formulate and implement policies integrating these determinants for holistic agrotourism development. These policies should underscore the significance of multi-stakeholder involvement, which includes local communities, businesses, and educational institutions. Moreover, these policies should cater to both environmental and social sustainability, ensuring that the growth of agrotourism does not detrimentally impact the rural environment and, instead, provides long-term benefits for the local community.

It is also recommended that governments and stakeholders prioritize capacity-building initiatives, such as educational programs and training, to enhance local communities' knowledge and skills in managing agrotourism activities. Financial support mechanisms should also be established to address capital and financial constraints faced by local communities.

This study, while providing insightful findings, acknowledges several limitations. Primarily, while yielding rich insights, its qualitative nature does not enable the establishment of causal relationships between factors and may miss some quantitative aspects for a broader perspective. Moreover, the study's focus is chiefly on internal and external agrotourism development factors, potentially overlooking unexplored influences such as geopolitical elements or global tourism trends. Additionally, the study represents a snapshot in time, implying that the relevance of its findings could vary due to the dynamic nature of agrotourism development.

Future research could address these constraints by adopting a mixed-methods approach and widening the investigatory scope. In order to capture the evolving nature of agrotourism, longitudinal studies could prove beneficial. Future explorations should also scrutinize the impact of policy implementation on agrotourism development trajectories, specifically how they may promote or impede stakeholder engagement and contribute to environmental and social sustainability. Emphasizing the social and environmental implications of agrotourism will also be crucial. Such in-depth examinations in the future would significantly contribute to maintaining the sustainability and mutual benefit of agrotourism in the long run.

**Author Contributions:** Conceptualization, Z.Z. and J.J.; methodology, J.J.; software, D.H. and F.R.A.; validation, E.R. and A.E.P.; formal analysis, Z.Z.; investigation, F.R.A.; resources, D.H.; data curation, D.H.; writing—original draft preparation, J.J.; writing—review and editing, Z.Z.; visualization, E.R.; supervision, A.E.P.; project administration, Z.Z.; funding acquisition, E.R. All authors have read and agreed to the published version of the manuscript.

**Funding:** This research and APC were funded by the Institution of Research and Community Services, Universitas Jambi. Skema Penelitian Kerjasama Nasional (Scheme of National Collaborative Research). Contract Number: 2166/UN21.11/01.05/SPK/2022 dated 2 June 2022.

**Informed Consent Statement:** Not applicable.

**Data Availability Statement:** The corresponding author [Z.Z.] of the present work is available for any information about data availability.

**Conflicts of Interest:** The authors declare no conflict of interest.

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
