# Peer review of "Understanding the Emergence of Rural Agrotourism: A Study of Influential Factors in Jambi Province, Indonesia"

_economies, doi:10.3390/economies11070180_

Round 1

Reviewer 1 Report

Interesting paper representing experience of agrotourism in Indonesia. The authors have used relevant information and literature sources. Some suggestions for improvements:

1) For the international readership, it would be good to add a map of Indonesia and particularly the sample territories. Also some general information about the agrotourism (statistics, its impact on national economy, etc.) in the country would be useful.

2) The methodology part must be improved greatly. Information about the research process, sampling, research ethics and data analysis must be provided. When and how the FGDs and interviews were organized? How many people were interviewed and how many FGDs were held? What were the main topics and questions discussed? How exactly triangulation was implemented?

3) From the current data analysis it is not clear, how the data particularly from the FGDs and the interviews was analyzed. Some quotes could help to better understand the data.

Good luck to the authors.

Author Response

Dear reviewer

I would like to express my deepest gratitude to the reviewers for their time and effort in reading and reviewing my article. Your valuable comments and suggestions have greatly contributed to improving the quality of the manuscript. Thank you for your expertise and dedication to maintaining the high standards of our scholarly community. Our response to your review is attached.

Response to Reviewer 1 Comments

Point 1: For the international readership, it would be good to add a map of Indonesia and particularly the sample territories. Also some general information about the agrotourism (statistics, its impact on national economy, etc.) in the country would be useful..

Response 1: Thank you for your suggestion. We have added a map of Indonesia and the research locations and general information about Indonesia's agrotourism in the introduction section..

Point 2: The methodology part must be improved greatly. Information about the research process, sampling, research ethics and data analysis must be provided. When and how the FGDs and interviews were organized? How many people were interviewed and how many FGDs were held? What were the main topics and questions discussed? How exactly triangulation was implemented?.

Response 2: Thank you for your advice. We have improved and detailed the Methodology section, including information about the research process, the number of FGD meetings, who was interviewed, topics of FGD and interviews, and the implementation of triangulation.

Point 3: From the current data analysis it is not clear, how the data particularly from the FGDs and the interviews was analyzed. Some quotes could help to better understand the data.

Response 3: Thank you. We have detailed in the methodology section the process of data analysis sourced from FGDs and interviews. Also, we have added several quotes from the interviews in the results and discussion section.

Reviewer 2 Report

Thank you for the opportunity to read this article. I found it interesting, but there are some issues that need to be addressed. 

First of all, from the introduction section, the aim of the paper is not so clear, that is, I was not convinced why this research was necessary. As authors said - many studies have explored the factors influencing the development of agro-tourism, but their reason was not convincing. Using a reason that "exploring several distinct locations rather than focusing on a single site" is not enough. That is why I do not see novelty in this research, especially contribution. 

Results and Discussion - Subsection 3.1. is more appropriate for the methodology section as a case study description, since these are not the results of research. Please, provide an adequate references for this subsection. 

Factors influencing Agrotourism Development - this section must be improved. We have results, but we do not know how we got them - through literature review or from the focus group discussion. Authors mentioned that they did in-depth interviews with a wide array of stakeholders, but where those results are? 

Is there anything different in your research from other research? What are the theoretical implications of the research? Why is this research significant when it confirmed everything that we already know regarding the factor that influenced development of agrotourism? 

What are limitations of this research? 

In this form, the paper has really low originality. 

Author Response

Dear reviewer

I would like to express my deepest gratitude to the reviewers for their time and effort in reading and reviewing my article. Your valuable comments and suggestions have greatly contributed to improving the quality of the manuscript. Thank you for your expertise and dedication to maintaining the high standards of our scholarly community. Our response to your review is attached.

Response to Reviewer 2 Comments

Point 1: First of all, from the introduction section, the aim of the paper is not so clear, that is, I was not convinced why this research was necessary. As authors said - many studies have explored the factors influencing the development of agro-tourism, but their reason was not convincing. Using a reason that "exploring several distinct locations rather than focusing on a single site" is not enough. That is why I do not see novelty in this research, especially contribution.

Response 1: Thank you. We have clarified the novelty of this research in the last three paragraphs of the introduction section.

"Previous research in this domain has indisputably established a robust foundation. However, most prior studies primarily investigated single or multiple locations sharing similar agrotourism characteristics. This approach resulted in individual studies identifying different and location-specific factors, leading to a relatively wide variation of identified elements. The absence of a common factor influencing the development of agrotourism villages inevitably hinders the formulation of effective policies applicable across diverse agrotourism village characteristics.

This study diverges from its predecessors by exploring several distinct locations rather than concentrating on a single site or multiple sites with similar agrotourism traits. The research scrutinizes four various agrotourism sites: those founded on forest resources, coastal and mangrove environments, horticultural plants, and tea plantations. The study aims to identify common determinants by examining these diverse agrotourism areas, which can be broadly applied across various agrotourism locations.

Theoretically, this approach is expected to find common factors influencing the de-velopment of agrotourism villages across varied village characteristics. Furthermore, building on this foundation of common factors, this study aims to practically contribute to formulating policies for developing agrotourism villages. The policies will generally apply across various village characteristics, enhancing their effectiveness and sustainability”

Point 2: Results and Discussion - Subsection 3.1. is more appropriate for the methodology section as a case study description, since these are not the results of research. Please, provide an adequate references for this subsection.

Response 2: Thank you for your suggestion. We have moved Subsection 3.1 to the methodology section. We have also cited several references, especially data sources, to strengthen the information.

Point 3: Factors influencing Agrotourism Development - this section must be improved. We have results, but we do not know how we got them - through literature review or from the focus group discussion. Authors mentioned that they did in-depth interviews with a wide array of stakeholders, but where those results are?.

Response 3: The analyzed data were derived from FGDs and in-depth interviews. We have improved the methodology section to clarify this.

Point 4: What are limitations of this research?.

Response 4: Thank you. We have added the limitations of this research in the conclusion section.  

"This study, while providing insightful findings, acknowledges several limitations. Primarily, while yielding rich insights, its qualitative nature doesn't enable establishing causal relationships between factors and may miss some quantitative aspects for a broader perspective. Moreover, the study's focus is chiefly on internal and external agrotourism development factors, potentially overlooking unexplored influences such as geopolitical elements or global tourism trends. Additionally, the study represents a snapshot in time, implying that the relevance of its findings could vary due to the dy-namic nature of agrotourism development.”

Reviewer 3 Report

Dear Authors

I hope this letter finds you well. I have carefully reviewed your research paper titled “Demystifying the Growth of Rural Agrotourism: A Study of Influential Factors in Jambi Province, Indonesia “, that was submitted to EJIHPE journal. I appreciate the effort and time you invested in conducting this study and preparing the manuscript for publication.

However, after a thorough evaluation, the paper has some minore issue as below:

·         Justify using qulitative research approch over the quntitative one

·         Practical and theoretical contributions are missing.

·         Please note that these comments and recommendations are intended to assist you in improving your research paper. I encourage you to carefully consider this feedback and address the mentioned concerns in your future revisions.

Dear Authors

I hope this letter finds you well. I have carefully reviewed your research paper titled “Demystifying the Growth of Rural Agrotourism: A Study of Influential Factors in Jambi Province, Indonesia “, that was submitted to EJIHPE journal. I appreciate the effort and time you invested in conducting this study and preparing the manuscript for publication.

However, after a thorough evaluation, the paper has some minore issue as below:

·         Justify using qulitative research approch over the quntitative one

·         Practical and theoretical contributions are missing.

·         Please note that these comments and recommendations are intended to assist you in improving your research paper. I encourage you to carefully consider this feedback and address the mentioned concerns in your future revisions.

Author Response

Dear reviewer

I would like to express my deepest gratitude to the reviewers for their time and effort in reading and reviewing my article. Your valuable comments and suggestions have greatly contributed to improving the quality of the manuscript. Thank you for your expertise and dedication to maintaining the high standards of our scholarly community. Our response to your review is attached.

Response to Reviewer 3 Comments

Point 1: Justify using qulitative research approch over the quntitative one.

Response 1: Thank you. Justification for using a qualitative approach has been added in the methodology section.

“The data were analyzed using a qualitative approach. Through this qualitative approach, it is hoped that the research can reveal an in-depth, detailed, and natural understanding of agrotourism in rural areas and provide an opportunity to uncover new findings or themes unrecognized or unmeasurable by quantitative methods.”

Point 2: Practical and theoretical contributions are missing.

Response 2: Thank you. It has been added in the last paragraph of the introduction.

“Theoretically, this approach is expected to find common factors influencing the development of agrotourism villages across varied village characteristics. Furthermore, building on this foundation of common factors, this study aims to practically contribute to formulating policies for developing agrotourism villages. The policies will generally apply across various village characteristics, enhancing their effectiveness and sustainability.”

Reviewer 4 Report

Congratulations on your article; it is clear and presented in a correct manner.  

However, I have only set of recommendations, namely:

a)       To rethink about the paper title “Demystifying the Growth of Rural Agrotourism…”.

Did you “demystify” the Growth of rural AgroTourism? In what sense? Please clarify. As you said (in the conclusion)  and many authors also underlined before : “it is a complex and multidimensional process”

b)     You considered “Natural tourism” and it’s correct. However, “Nature Tourism” is more frequent in the academic literature.

c)      In the lines 103-104 you said:  “This study diverges from its predecessors by exploring several distinct locations rather than focusing on a single site or multiple sites with similar agro-tourism traits”.  In lines 122-114 you said something similar. Please try not to rephrase.

d)     Please uniformize (in the text): “agrotourism” or “agro-tourism”

e)      Regarding the section “Methodology”, please give us more information about:

·        How many Focus Group meetings did you do? (Focus Group for each Village? Or For all the stakeholders?) and when?

·        How many interviews?

·        How many participants did you have in each Focus Group Meeting?

·        Stakeholders' interviews are a critical part of the article, as they allow you to gain a deeper understanding of agrotourism. The interviews were applied to which stakeholders and why? Please justify.

·        When (before, after or during the focus groups) did you interview them? 

f)       The point “3.1. Sample Village Profile” underlines the different characteristics of the 4 villages. How did you do the characterization? Through field observation? Through academic sources? Please clarify.

g)      In the context of villages characterization, it is important to underline that a tourist product has a set of elements (for instance: attributes of the place, activities of the place, travel and tourism industry companies, and additional elements (such as: accessibilities, guides, tourism office, etc). Simultaneously, in the characterization of the villages, authors considered the tourism product stage regarding 2 villages, but regarding the other ones they didn’t inform us. Are the villages in the emergence stage of the tourism product?

h)     In the conclusion it could be important to underline which are the development factors common to all the 4 villages and the factors that put in evidence asymmetries.  

Author Response

Dear reviewer

I would like to express my deepest gratitude to the reviewers for their time and effort in reading and reviewing my article. Your valuable comments and suggestions have greatly contributed to improving the quality of the manuscript. Thank you for your expertise and dedication to maintaining the high standards of our scholarly community. Our response to your review is attached.

Response to Reviewer 4 Comments

Point 1: To rethink about the paper title “Demystifying the Growth of Rural Agrotourism…”.

Did you “demystify” the Growth of rural AgroTourism? In what sense? Please clarify. As you said (in the conclusion)  and many authors also underlined before : “it is a complex and multidimensional process..

Response 1: Thank you for your valuable feedback. After discussing with the author team, we have revised the manuscript title to “Understanding the Emergence of Rural Agrotourism: A Study of Influential Factors in Jambi Province, Indonesia”

Point 2: You considered “Natural tourism” and it’s correct. However, “Nature Tourism” is more frequent in the academic literature.

Response 2: Thank you for the correction. We have replaced all instances of "natural tourism" with "nature tourism".

Point 3: In the lines 103-104 you said:  “This study diverges from its predecessors by exploring several distinct locations rather than focusing on a single site or multiple sites with similar agro-tourism traits”.  In lines 122-114 you said something similar. Please try not to rephrase.

Response 3: Thank you for the correction. We have revised this part to avoid repetition and now it is integrated into the last three paragraphs of the introduction section.

“Previous research in this domain has indisputably established a robust foundation. However, most prior studies primarily investigated single or multiple locations sharing similar agrotourism characteristics. This approach resulted in individual studies identifying different and location-specific factors, leading to a relatively wide variation of identified elements. The absence of a common factor influencing the development of agrotourism villages inevitably hinders the formulation of effective policies applicable across diverse agrotourism village characteristics.

This study diverges from its predecessors by exploring several distinct locations rather than concentrating on a single site or multiple sites with similar agrotourism traits. The research scrutinizes four various agrotourism sites: those founded on forest resources, coastal and mangrove environments, horticultural plants, and tea plantations. The study aims to identify common determinants by examining these diverse agrotourism areas, which can be broadly applied across various agrotourism locations.

Theoretically, this approach is expected to find common factors influencing the de-velopment of agrotourism villages across varied village characteristics. Furthermore, building on this foundation of common factors, this study aims to practically contribute to formulating policies for developing agrotourism villages. The policies will generally apply across various village characteristics, enhancing their effectiveness and sustainability.”

Point 4: Please uniformize (in the text): “agrotourism” or “agro-tourism”.

Response 4: Thank you for the correction. We have replaced all instances of "agro-tourism" with "agrotourism".

Point 5: Regarding the section “Methodology”, please give us more information about:

  • How many Focus Group meetings did you do? (Focus Group for each Village? Or For all the stakeholders?) and when?
  • How many interviews?
  • How many participants did you have in each Focus Group Meeting?
  • Stakeholders' interviews are a critical part of the article, as they allow you to gain a deeper understanding of agrotourism. The interviews were applied to which stakeholders and why? Please justify.
  • When (before, after or during the focus groups) did you interview them?.

Response 5: Thank you for your suggestions. We have revised and detailed the Methodology section to provide information about the research process, the number of FGD meetings, interviewees, and the topics of the FGD and interviews.

Point 6: The point “3.1. Sample Village Profile” underlines the different characteristics of the 4 villages. How did you do the characterization? Through field observation? Through academic sources? Please clarify..

Response 6: Thank you. The information about the sample village profiles came from field observations, interviews with village administrators, and documents available from the village government. We have added this explanation at the beginning of the sample village profile section..

“The following section provides a concise profile of the four sampled villages, the information for which is derived from observational studies, interviews conducted with the village administrative staff, and available documents within the village governance”.

Point 7: In the context of villages characterization, it is important to underline that a tourist product has a set of elements (for instance: attributes of the place, activities of the place, travel and tourism industry companies, and additional elements (such as: accessibilities, guides, tourism office, etc). Simultaneously, in the characterization of the villages, authors considered the tourism product stage regarding 2 villages, but regarding the other ones they didn’t inform us. Are the villages in the emergence stage of the tourism product?.

Response 7: Thank you. We have revised the writing structure in the sample village profiles section. We have also added additional information that was previously missing.

Point 8: In the conclusion it could be important to underline which are the development factors common to all the 4 villages and the factors that put in evidence asymmetries.

Response 8: Thank you, we have added this in the conclusion. 

“The development of agrotourism is a multifaceted and intricate process necessitating coordination and collaboration among a range of stakeholders, encompassing local governments, communities, and business entities. This research identifies six common factors that impact the evolution of agrotourist villages, namely, nature tourism potential, agroproducts, heritage, infrastructure, community and institution, and technology innovation.

The first factor, nature tourism potential, indicates the importance of a region's natural attributes and assets. These could include unique landscapes, flora and fauna, or other natural features that draw tourists. Agroproducts the second factor, refer to the agricultural goods and experiences that tourists can offer. This could range from traditional farming practices and local crops to culinary experiences and locally made products.

Heritage, the third factor, underscores the cultural significance of an area and its people. This could involve historical sites, traditional arts and crafts, folklore, festivals, and any aspect that reflects the community's heritage. Infrastructure, as the fourth factor, is crucial as it includes not just transportation and accessibility but also the availability of suitable accommodation, amenities, and other services necessary for tourism.

The fifth factor, community and institution, underscores the importance of social structures, local governance, and community engagement in successful agrotourism. A supportive local community and effective institutions can greatly facilitate agrotourism initiatives. Finally, the sixth factor, technology innovation, indicates the increasingly important role that modern technologies, such as digital marketing, online booking systems, and other innovative farming or tourism-related technologies, play in contemporary agrotourism development”.

Reviewer 5 Report

Dear authors,

Very interesting study. Only some aspects must be reviewed:

-In conclusion part your proposals for the future, it would be appropriate.

- In research part maybe establishing a hierarchy of influencing factors it would be necessary as conclusion of the research!

- Specifying the hypothesis of the research must be made in the introduction part (I have seen that it is mentioned the purpose, and this purpose could be rephrased so as to become the research hypothesis)!

- Maybe some minor corrections about final form or spelling can be made! E.g you have in the paper agro-tourism and in some places agrotourism...you must use the same term.

Overal it is a good research!

Author Response

Dear reviewer

I would like to express my deepest gratitude to the reviewers for their time and effort in reading and reviewing my article. Your valuable comments and suggestions have greatly contributed to improving the quality of the manuscript. Thank you for your expertise and dedication to maintaining the high standards of our scholarly community. Our response to your review is attached.

Response to Reviewer 5 Comments

Point 1: In conclusion part your proposals for the future, it would be appropriate.

Response 1: Thank you. In the conclusion section, we have revised and added proposals for the future.

“Future research could address these constraints by adopting a mixed-methods approach and widening the investigatory scope. To capture the evolving nature of agrotourism, longitudinal studies could prove beneficial. Future explorations should also scrutinize the impact of policy implementation on agrotourism development trajectories, specifically how they may promote or impede stakeholder engagement and contribute to environmental and social sustainability. Emphasizing the social and environmental implications of agrotourism will also be crucial. Such in-depth examinations in the future would significantly contribute to maintaining the sustainability and mutual benefit of agrotourism in the long run.”

Point 2: In research part maybe establishing a hierarchy of influencing factors it would be necessary as conclusion of the research!.

Response 2: Thank you for your suggestion. In the results and discussion section, we have added a table with a hierarchy of influencing factors and their explanation..

Point 3: Specifying the hypothesis of the research must be made in the introduction part (I have seen that it is mentioned the purpose, and this purpose could be rephrased so as to become the research hypothesis)!.

Response 3: Thank you. Our research uses a qualitative approach, emphasizing exploration and in-depth understanding of certain phenomena rather than testing pre-determined hypotheses or predictions. Our research is guided by the research objective: "The study aims to identify common determinants by examining these diverse agrotourism areas, which can be broadly applied across various agrotourism locations.".

Point 4: Maybe some minor corrections about final form or spelling can be made! E.g you have in the paper agro-tourism and in some places agrotourism...you must use the same term.

Response 4: Thank you for your correction. We have changed all words in the paper to 'agrotourism'.

Round 2

Reviewer 2 Report

Authors provided much improved version of the paper and I am quite satisfied with the added changes. I recommend publishing this paper.